# The Poplar (*Populus trichocarpa*) Dehydrin Gene *PtrDHN-3* Enhances Tolerance to Salt Stress in Arabidopsis

**DOI:** 10.3390/plants11202700

**Published:** 2022-10-13

**Authors:** Meiqi Zhou, Nafei Peng, Chuanping Yang, Chao Wang

**Affiliations:** State Key Laboratory of Tree Genetics and Breeding, Northeast Forestry University, Harbin 150040, China

**Keywords:** *Populus trichocarpa*, *PtrDHN-3*, salt stress, ROS-scavenging capability

## Abstract

Dehydrin (DHN), a member of the late embryogenesis abundant protein (LEA) family, was recently found to play a role in physiological responses to salt and drought stress. In this study, we identified and cloned the *PtrDHN-3* gene from *Populus trichocarpa*. The PtrDHN-3 protein encoded 226 amino acids, having a molecular weight of 25.78 KDa and an isoelectric point of 5.18. It was identified as a SKn-type DHN and was clustered with other resistance-related DHN proteins. Real-time fluorescent quantitative PCR showed that transcription levels of *PtrDHN-3* were induced by mannitol stress, and more significantly by salt stress. Meanwhile, in a yeast transgenic assay, salt tolerance increased in the *PtrDHN-3* transgenic yeast, while the germination rate, fresh weight and chlorophyll content increased in *PtrDHN-3*-overexpressing transgenic *Arabidopsis* plants (OE) under salt stress. Significant increases in expression levels of six antioxidant enzymes genes, and SOD and POD enzyme activity was also observed in the OE lines, resulting in a decrease in O_2_^-^ and H_2_O_2_ accumulation. The proline content also increased significantly compared with the wild-type, along with expression of proline synthesis-related genes *P5CS1* and *P5CS2*. These findings suggest that *PtrDHN-3* plays an important role in salt resistance in plants.

## 1. Introduction

Extremely harsh environmental conditions, such as high and low temperatures, saline-alkali conditions and drought, have a severe effect on plant growth and development, and are the main causes of plant mortality worldwide [1]. To survive adversity, plants have therefore evolved a series of mechanisms aimed at resisting stress. For example, salinity, high and low temperatures, drought, and other abiotic stresses cause water stress within the plant, in turn inducing a series of physiological reactions, such as increases in the contents of proline, soluble protein, carbohydrates, and organic acids, and a loss of lipid membranes [2]. These reactions are also thought to alter or induce the production of additional proteins, and all of these metabolic changes are regulated by gene expression [3,4]. Identification of the genes related to abiotic stress is therefore an important focus of research into the molecular mechanisms of plant stress resistance.

Dehydrins (DHNs) are members of the late embryogenesis abundant protein (LEA) family [5] which accumulate in the late stages of embryogenesis when water content in seeds declines, or in response to various stressors. DHN is clustered into the group II of highly hydrophilic proteins and has been discovered in a large number of plant species [6,7]. DHN usually contains a highly conserved N-terminal Y segment [T/V]D[E/Q]YGNP), a C-terminal K segment (EKKGIMDKIKEKLPG), and a central S segment (serine-track) [8,9]. Studies indicate that the K segments are the functional core parts of DHNs that mediate cellular stress tolerance or maintain enzyme activity [10]. S segments are generally thought to affect the localization or function of DHNs by phosphorylation [11], while the Y-segment is composed of aromatic conserved tyrosine residues the function of which remains unclear [12,13]. According to the number and position of these conserved domains, they can be divided into five different types: Kn, KnS, YnKn, SKn, and YnSKn. Kn dehydrogenase is induced by low temperatures, drought, and abscisic acid [14]; SKn-type DHN is induced by various abiotic stresses [15,16,17]; and the K-fragment of KnS-type DHN is induced by low temperatures and drought [18].

DHN is widely involved in the physiological responses to abiotic stress, playing an important role in improving resistance. Numerous in vitro studies have provided evidence for the potential role of different DHNs in plants. Among the various biochemical activities proposed for these proteins, many suggest a function as membrane or protein stabilizers [19], as radical scavengers protecting lipids against peroxidation, or as chaperones preventing stress-induced aggregation of enzymes or proteins [20]. In plum (*Prunus mume*), *DHN* gene expression was induced by ABA (abscisic acid), SA (salicylic acid), high and low temperatures, PEG (polyethylene glycol), and Salt. Meanwhile, in recombinant *Escherichia coli*, overexpression of *PmLEA10*, *PmLEA19*, *PmLEA20*, and *PmLEA29* were found to improve permeability and resistance to freezing [21]. Furthermore, compared with wild-type wheat, transgenic plants containing the YnSKn-type DHN gene *WZY2* had a lower water content and higher malondialdehyde (MDA) content [22]. Tomato (*Solanum lycopersicum* L.) plants transformed with the yacon DHN gene *SiDHN* also showed increases in chlorophyll a and b, relative water, and carotenoid contents, and a significant decrease in H_2_O_2_ and O^2-^ [23]. In Arabidopsis, the KS-type DHN *AtHIRD11* inhibited the production of hydrogen peroxide and hydroxyl free radicals in the copper ascorbate system [24], while knock-out of DHN genes *Gh_A05G1554* (*GhDHN_03*) and *Gh_D05G1729* (*GhDHN_04*) reduced the ability of cotton plants to tolerate osmotic pressure and salt stress [25]. Furthermore, mutant plants obtained via knock-out of *PpDHNA*, a *DHN* gene isolated from the moss *Physcomitrella patens*, expressed severely impaired growth under salt and osmotic stress [26]. The important protective role of DHN during cell dehydration has also been shown.

In this study, we identified and cloned the *PtrDHN-3* gene from *Populus trichocarpa* in order to clarify its role in abiotic stress responses. In doing so, its function, and related physiological and molecular pathways under salt tolerance were determined in transgenic yeast and Arabidopsis assays. The results provide a foundation for future research into the functions of DHN in plant stress resistance.

## 2. Results 

### 2.1. Phylogenetic Tree Analysis of PtrDHN-3

The full-length CDS of *PtrDHN-3* was successfully cloned from *P. trichocarpa*. The PtrDHN-3 protein encoded 226 amino acids, with a molecular weight of 25.78KDa and an isoelectric point of 5.18. A phylogenetic tree was then constructed with other proteins of known function within different DHN subfamilies from other species (Figure 1A). PtrDHN-3 was homologous with DHN proteins possessing stress-resistance functions and was closest with IpDHN [27]. Furthermore, multiple sequence analysis of DHN proteins (OesDHN, StDHN and IpDHN) clustered with PtrDHN-3 revealed that the PtrDHN-3 protein contains three K segments and one S segment, representing a SK3 DHN belonging to the SKn-type subfamily (Figure 1B).

### 2.2. Characterization of Cis Elements in the PtrDHN-3 Gene Promoter

The 1500-bp upstream sequence of the initiator codon ATG in the *PtrDHN-3* gene was selected for analysis of *cis*-acting elements in the gene promoter. Prediction analysis showed a number of *cis*-acting elements related to stress responses (Figure 2), such as an MYB binding site (MBS), ABA response element (ABRE), MYC recognition site, DRE recognition site, and general transcription factor binding sites such as TATA box.

### 2.3. Analysis of PtrDHN-3 Expression under Abiotic Stress

To determine the response of *PtrDHN-3* to abiotic stress, qRT-PCR analysis was used to obtain expression profiles under respective treatment with 100 mM NaCl and 200 mM mannitol. Under mannitol stress (Figure 3A), the transcripts of *PtrDHN-3* were first induced in the roots at 6 h followed by the stems and leaves at 12 h, with a peaking at 12 h in all tissues. Peak transcription levels in the leaves were 14 times greater than those of the control. These results indicate that *PtrDHN-3* plays a role in the response to mannitol-induced drought stress. Meanwhile, treatment with NaCl caused a rapid increase in transcription levels of PtrDHN-3 in the roots, stems and leaves, with a notable increase in the leaves. Levels were highest at 24 h, at 41 times that of the control (Figure 3B). Since the response to salt stress was greater than that to drought, subsequent analyses focused on the function of *PtrDHN*-3 in salt stress.

### 2.4. PtrDHN-3 Improves Salt Tolerance in Yeast

To verify the potential function of *PtrDHN-3* in salt stress, pYES2-*PtrDHN-3* recombinant plasmids were transferred into the INVSC1 yeast strain. An empty pYES2 vector was used as a control. Under control conditions, there were no obvious differences in growth between the control and *PtrDHN-3* transgenic cells. Meanwhile, under NaCl stress, growth of the *PtrDHN-3* transgenic yeast was greater than that of the control at each dilution (10, 100, and 1000). Notably, the survival rate of transgenic yeast cells diluted 100-fold was much greater than that of control (Figure 4). 

### 2.5. Overexpression of PtrDHN-3 Increases Salt Tolerance in Arabidopsis

To confirm the role of *PtrDHN-3* in salt tolerance, *PtrDHN-3*-overexpressing *Arabidopsis* plants (OE) were constructed. *Agrobacterium*-mediated transformation of *PtrDHN-3* in ten *Arabidopsis* lines was confirmed by PCR (Appendix A). Two T3 homozygous lines (OE-8 and OE-10) were then randomly selected for analysis of salt tolerance. When grown on 1/2 MS medium, no obvious differences in germination rates were observed between the transgenic and WT lines. However, on 1/2 MS medium containing 100 mM NaCl, germination rates were about three times greater in the transgenic lines than the WT (Figure 5A,B). Fresh weights were also significantly greater in the OE lines compared to the WT under salt stress (Figure 5C). Under control conditions, no significant differences in growth phenotypes were observed between the transgenic and WT plants; however, under salt stress, growth was better in the transgenic compared to the WT plants, and rosette leaves were healthier looking (Figure 5D). Chlorophyll contents decreased under salt stress in both the OE and WT plants; however, the decrease was greater in the WT. 

### 2.6. PtrDHN-3 Scavenges ROS

Under normal growth conditions, Diaminobenzene (DAB) and Nitrotetrazolium Blue chloride (NBT) staining revealed similar accumulation of H_2_O_2_ and O_2_^·^^-^ between the transgenic and WT Arabidopsis plants (Figure 6A). However, under salt stress conditions, accumulation of H_2_O_2_ and O_2_^-^ decreased in the transgenic compared with WT plants, especially at 12 h, and blue and brown staining were detected in the leaves. 

Analysis of antioxidant enzymes activity showed that SOD and POD activities were significantly higher in the transgenic compared to WT plants under salt stress (Figure 6B,C). Meanwhile, qRT-PCR analysis of *SOD* and *POD* gene expression revealed no significant differences between the OE and WT plants under non-stress conditions (Figure 6D), but a significant increase under salt stress in the transgenic plants (Figure 6E). 

### 2.7. PtrDHN-3 Improves Biosynthesis of Proline under Salt Stress

Studies have shown that proline content can be used to measure salt tolerance in plants [28]. To determine whether *PtrDHN-3* induces proline biosynthesis, proline contents were measured in the transgenic and WT Arabidopsis. Under non-stress conditions, proline contents were similar between the transgenic and WT plants; however, under salt stress, contents increased significantly in the transgenic plants (Figure 7A). Expression of proline biosynthesis-related genes *P5CS1* and *P5CS2* has also been studied in relation to salt stress [29]. In this study, no obvious differences were observed between the transgenic and WT lines under normal conditions; however, under salt stress, transcript levels of both genes increased in the transgenic plants (Figure 7B). 

## 3. Discussion

Considerable research has been carried out into the role of *DHN* genes in cold stress resistance [20]. Meanwhile, in more recent years, *DHN* has also been found to play a role in drought resistance and salt tolerance [23,25]. However, few studies have examined the role of this gene in trees. A previous study in poplar (*Populus trichocarp*) revealed that *PtrDHN* genes are related to abiotic stress responses [30]. Therefore, in this study, the function of *PtrDHN3* in response to stress was investigated further.

Phylogenetic tree analysis showed that PtrDHN-3 is clustered with a number of stress-resistant genes including IpDHN, OesDHN, StDHN and CcDHN (Figure 1A), the functions of which in relation to abiotic stress have been previously identified [31,32,33]. Meanwhile, multiple sequence alignment revealed that PtrDHN-3 is a SKn-type DHN protein (Figure 1B). The functions of different types of DHN proteins differ, together with their responses to stress. For example, studies have shown that SKn-type DHNs are a type of acid DHN that are induced by multiple stresses [15,16,17]. In this study, analysis of the promoter region of the *PtrDHN*-*3* gene revealed a variety of elements related to stress, such as an MYB binding site (MBS), ABA response element (ABRE), MYC recognition site and DRE recognition site (Figure 2). These sequence characteristics therefore suggest that PtrDHN-3 is involved in abiotic stress responses in poplar. 

In general, genes related to abiotic stress resistance are induced under stress conditions. In order to verify whether this is the case with *PtrDHN-3*, responses to salt and drought stress were examined. Transcript levels of *PtrDHN-3* were induced under both mannitol and NaCl treatment; however, the response was greater under salt stress, especially in the leaves (Figure 3), suggesting an important role in salt stress tolerance. This was subsequently confirmed via histochemical staining, and analyses of SOD activity, POD activity, and chlorophyll and proline contents. 

Yeast is an important tool for screening genes involved in abiotic stress resistance. For example, transference of *ThvhAC1* gene from *Tamarix hispida* into yeast cells improved tolerance and survival of transgenic yeast under abiotic stress [34], while overexpression of *TdSHN1* (*Durum wheat* gene) improved tolerance to Cd, Cu, and Zn stress in transgenic yeast [35]. In this study, *PtrDHN-3* improved tolerance to salt stress in the transformed yeast cells (Figure 4), although there was no obvious difference under mannitol-induced drought stress (data not shown).

To verify the function of *PtrDHN-3* in salt tolerance, *PtrDHN-3* overexpression was induced in Arabidopsis. Analysis showed that under NaCl stress conditions, the transgenic *Arabidopsis* plants had a greater germination rate, fresh weight and chlorophyll content than the WT. Various germination tests and phenotype analyses can be used to show the role of resistance genes in salt tolerance [36]. The present results suggest that *PtrDHN-3* functions in salt tolerance by improving germination and reducing tissue dehydration and photosynthetic damage under stress conditions (Figure 5).

When subjected to abiotic stress, plants produce a large amount of ROS, thereby inducing oxidative harm. Wheat *DHN* was found to improve salt tolerance in transgenic *Arabidopsis thaliana* by eliminating ROS [37], while cellular accumulation of *IpDHN* was found to improve salt tolerance in transgenic Arabidopsis, possibly by activating the oxygen-scavenging system [27]. DAB and NBT staining are commonly used to determine the contents of H_2_O_2_ and O_2_^-^ [38], and in this study, were used to identify the function of *PtrDHN3* in ROS scavenging (Figure 6A). The results showed that overexpression of *PtrDHN3* reduced the accumulation of both H_2_O_2_ and O_2_^-^ under salt stress conditions, suggesting that *PtrDHN3* improves salt tolerance by reducing ROS accumulation. Plants possess an antioxidant enzyme system that protects cells from oxidative damage by scavenging ROS. SOD and POD play an important role in ROS detoxification and are therefore crucial factors in abiotic stress resistance. In this study, a significant increase in SOD and POD activity were observed under salt stress in the *PtrDHN-3* transgenic plants (Figure 6B,C), thereby functioning to reduce ROS accumulation. Expression levels of *SOD* and *POD* genes were also examined, revealing a significant increase in the transgenic component compared to WT plants under salt stress (Figure 6E). Consistent with this, previous knock-outs of *CaDHN4* and *CaDHN5* inhibited expression of manganese superoxide dismutase (MnSOD) and *peroxidase* (POD) genes, causing an increase in severe wilting and ROS accumulation in the leaves of Arabidopsis [39,40]. Taken together, these results suggest that *PtrDHN-3* enhances salt tolerance in the transgenic plants by inducing the expression of *SOD* and *POD*, improving the activities of SOD and POD, and thereby reducing ROS accumulation. 

Plants often accumulate proline under abiotic stress conditions, and proline is therefore thought to play an important role in plant cell adaptation to osmotic stress [41]. We therefore determined the expression of two important proline synthesis-related genes, P5CS1 and *P5CS2*, as well as the content of proline in transgenic *Arabidopsis* (Figure 7). Under salt stress conditions, *PtrDHN-3* induced an increase in the expression of *P5CS1* and *P5CS2*, and improved proline biosynthesis in the transgenic compared to WT plants. These findings suggest that overexpression of *PtrDHN-3* improves tolerance to salt stress by inducing the synthesis of proline. Consistent with this, the role of DHN in regulating proline synthesis to confer salt tolerance has also been shown in Australian wild rice [42]. 

## 4. Materials and Methods

### 4.1. Populus Trichocarpa Cultivation and Stress Treatment 

Four-week-old tissue culture seedlings of *P. trichocarpa* were transplanted into pots and cultivated in a greenhouse for two months under 14 h light/10 h dark, 70–75% relative humidity, and an indoor temperature of 25 °C. Two-month-old seedlings were then treated with 100 mM NaCl or 200 mM mannitol for 6, 12, 36, 24, 48, or 72 h. Well-watered seedlings were used as a control. Root, stem, and leaf samples from at least three seedlings per treatment were then harvested and pooled for real-time quantitative RT-PCR (qRT-PCR) analysis.

### 4.2. Cloning and Sequence Analysis of PtrDHN-3

Nucleic acid and amino acid sequences of *PtrDHN-3* were retrieved from the poplar genome database (https://phytozome.jgi.doe.gov/ (accessed on 15 November 2019)). Using *P. trichocarpa* cDNA as a template, the CDS sequence of *PtrDHN-3* was then amplified according to the full-length *PtrDHN-3* ORF sequence (GenBank accession no.: XM_002307732.1/Potri.005G248100.1) using forward and reverse primers 5′-ATGGCTGAGGAAAACAAGAGC-3′ and 5′-CTACTGGGAAGCACTCTCCT-3′, respectively. A phylogenetic tree was constructed using MEGA5 software [43], protein sequences were aligned using ClustalW2 (http://www.ebi.ac.uk/Tools/msa/clustalw2/ (accessed on 15 November 2019)), and the -1.5-kb upstream promoter sequence of the *PtrDHN-3* gene was searched using the phytozome database. plantCARE software (http://bioinformatics.psb.ugent.be/webtools/plantcare/html/ (accessed on 15 November 2019)) was used to analyze the *cis*-acting element in the promoter sequence of *PtrDHN-3* [44].

### 4.3. RNA Isolation and qRT-PCR

*PtrDHN-3* RNA was isolated using pBIOZOL Plant Total RNA extraction reagent (Bioflux, Beijing, China). cDNA was then synthesized from the total RNA using a Reverse Transcription Kit (Takara Japan, Osaka, Japan). 

Resistance-related gene sequences in Arabidopsis (*SOD1* (NM_100757), *SOD2* (NM_101123), *SOD3* (NM_128379), *POD1* (NM_102257), *POD2* (NM_127371), *POD3* (NM_127372), *P5CS1* (NM_129539) and *P5CS2* (NM_115419)) were download from https://www.arabidopsis.org/ (accessed on 15 November 2020). *PtrActin* (Potri.001G453600) and *AtActin* (At5G09810) were used as internal controls in *P. trichocarpa* and *Arabidopsis*, respectively. All primers used for qRT-PCR are shown in Appendix A.

Takara quantitative PCR enzyme (Takara Japan) was used for the qRT-PCR reactions, all of which were performed in triplicate as follows: 94 °C for 30 s followed by 45 cycles of 94 °C for 12 s, 58 °C for 30 s and 72 °C for 45 s. Detection was carried out using the Roche LightCycler 480 II (Jena q Tower 3G, Jena, Germany), and relative abundance was determined based on the 2^−ΔΔCT^ method [45]. All experiments were conducted with three biological replicates.

### 4.4. Salt Stress Tolerance Assays in Yeast

Analysis of salt tolerance in yeast was carried out using a yeast expression vector. PCR primers were designed based on the CDS sequence of the *PtrDHN-3* gene as follows: forward primer 5’-ACTCACTATAGGGAATATTA ATGGCTGAGGAAAACAAGAGC-3’ and reverse primer 5’-TAATTACATGATGCGGCCCTCTACTGGGAAGCACTCTCCT-3’. The *PtrDHN-3* gene was transferred into the yeast expression vector pYES2 (Carlsbad, CA, USA) using the Infusion enzyme (Zhongmei Taihe, c5891-50, Taihe, China). The constructed expression vector pYES2-*PtrDHN-3* was then transformed into INVSC1 yeast cells, with an empty pYES2 plasmid used as a control. The transformed cells were cultivated on an SC-U medium containing 2% glucose in an incubator at 30 °C for three days. Positive clones were then identified by PCR detection. Single-clone cells containing the recombinant DNA (pYES2-PtrDHN-3) and control vector cells were then incubated in an SC-U medium with shaking until an OD of 0.6 was reached. Yeast cells were then diluted 10-, 100- and 1000-fold and spotted on an SC-U medium. For the salt tolerance assay, yeast cells were re-suspended in 3 M NaCl for 24 h then spread on an an SC-U medium and inverted in an incubator at 30 °C for three days. Images were then obtained (Canon EOS 7D MARKII, Japan). As a control, the cultured yeast cells were spotted on an SC-U medium without NaCl treatment.

### 4.5. Construction of Transgenic Arabidopsis

Arabidopsis seeds (Col-0) were grown in grass carbon soil: vermiculite: perlite at a volume: volume ratio of 5: 3: 2, then placed in a cultivation room at 22~24 ℃ under a light/dark photoperiod of 16 h/8 h, and a light intensity of 80~120 μmol∙m^−2^∙s^−1^. The CDS sequence of *PtrDHN-3* was constructed in a pROKII vector (35S:: PtrDHN-3) [46] using PCR with forward primer 5’- CTCTAGAGGATCCCATGGCTGAGGAAAACAAGAGC -3’ and reverse primer 5’- TCGAGCTCGGTACCCCTACTGGGAAGCACTCTCCT -3’. The constructed 35S: *PtrDHN-3* plasmid was then transformed into *Agrobacterium* strain EHA105 for *Arabidopsis* infiltration. Transgenic lines were developed using the floral dipping method and screened in 1/2 MS selection medium with 50 mg/L kanamycin. PCR analysis was then carried out using forward primer 5’-AGACGTTCCAACCACGTCTT-3’ and reverse primer 5’-CCAGTGAATTCCCGATCTAG-3’. T3-generation homozygous *PtrDHN-3* transgenic lines were used for further experiments. 

### 4.6. Salt Stress Tolerance Assays in Arabidopsis

For analysis of salt stress tolerance, seeds of two transgenic Arabidopsis lines (OE-8, OE-10) and wild-type (WT) plants were sown on a 1/2 MS solid medium containing 100 mM NaCl. The germination rate was then observed after 10–14 days. Seeds of over-expressing T3 homozygous and control Arabidopsis lines were then sown on fresh 1/2 MS medium and observed for five days for germination. They were then transferred to 1/2 MS medium plates with or without (control) 100 mM NaCl and cultured at 22℃ with a photoperiod of 16 h light and a photon flux of 80–120 μmol∙m^−2^∙s^−1^. After 7-day continuous cultivation, fresh weights were then measured. Seedlings of 4-week-old T3 transgenic and WT Arabidopsis lines grown in pots were also treated with 100 mM NaCl solution for 10 days. Well-watered seedlings were used as a control. The planting method, nutrient substrate and cultivation environment were as described in Section 4.5. The phenotypes of the seedlings were then photographed with Canon EOS 7D MARKII, and contents of chlorophyll and proline in the leaves were determined. Chlorophyll contents was determined according to the description of Lightenthaler (1987) [47]. The samples were fully ground with liquid nitrogen and treated with acetone. The optical densities of supernate at wavelengths of 663 nm, 645 nm, 652 nm and 470 nm were measured. For proline assay, the ground samples were assayed by a colorimetric assay kit based on the ninhydrin reaction (Jiancheng Bio. Co., Nanjing, China) and the reaction product was quantified by its absorbance at 520 nm, calibrated with proline standards).

### 4.7. Histochemical Detection of Reactive Oxygen Species (ROS)

Leaves of 4-week-old T3 transgenic and WT Arabidopsis lines were sampled at 0, 3, 6 and 12 h after salt treatment. They were then stained with NBT (Nitrotetrazolium Blue chloride) for detection of H_2_O_2_ and with DAB (Diaminobenzene) for detection of superoxide (O_2˙_^−^) [48].

### 4.8. Analysis of Antioxidant Enzyme Activity

Leaves of 4-week-old T3 transgenic and WT Arabidopsis lines were sampled after 12 h salt treatment. Stress-treated samples were collected for analysis of antioxidant enzyme activity. Superoxide dismutase (SOD) and peroxidase (POD) activities were determined using corresponding kits based on the manufacturers’ instructions (Nanjing Jiancheng, Nanjing, China). Samples were obtained from at least nine seedlings per line, and all experiments were repeated three times.

### 4.9. Statistical Analyses

Statistical analyses were carried out using SPSS software version 20.0. Mean ± standard deviations (SD) were obtained from the average of three biological replicates. Data were compared using Student’s *t* test, and differences were considered significant at *p* < 0.05.

## 5. Conclusions

The DHN gene *PtrDHN-3* was successfully cloned from *P. trichocarpa.* Expression of *PtrDHN-3* was subsequently found to be induced by NaCl, suggesting a role in the response to salt stress. *PtrDHN*-*3* also improved tolerance to NaCl stress in transgenic yeast, as well as improving SOD and POD activity and reducing the accumulation of H_2_O_2_ and O_2_^−^ in transgenic *Arabidopsis*, preventing ROS oxidative harm and promoting osmotic regulation via accumulation of proline. Taken together, these findings suggest that *PtrDHN-3* plays an important physiological role in resistance to salt stress in plants, providing a foundation for future research into the functions of DHN in stress resistance.

## Figures and Tables

**Figure 1 plants-11-02700-f001:**
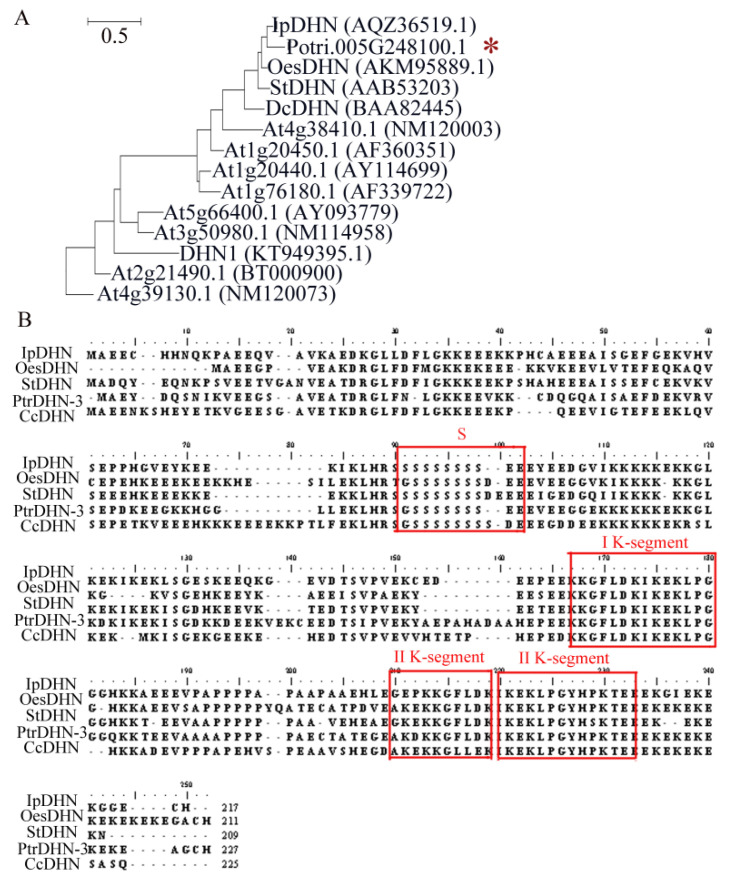
(**A**) Phylogenetic relationships between the PtrDHN-3 (Potri.005G248100.1) protein and dehydrins (DHNs) from other plant species. The molecular phylogeny was constructed from complete protein sequence alignment using the neighbor-joining method with ClustalW. The scale bar indicates the average number of amino acid substitutions per site. “*” Indicates the position of PtrDHN-3 in the phylogenetic tree. (**B**) Sequence analysis of the PtrDHN-3 protein. Multiple sequence alignment of the PtrDHN-3 protein with IpDHN (*Ipomoea pes-caprae* L. AQZ36519.1), OesDHN (*Olea europaea* L. subsp. europaea, var. Sylvestris, AKM95889.1), StDHN (*Solanum tuberosum* L. AAB53203), CcDHN (Coffea canephora Pierre ex Forehn. ABC68275) and DcDHN (*Daucus carota* L. BAA82445). Red squares indicate conserved amino acid sequences of S-and K-segment motifs. K segments include ⅠK segment: EKKGIMDIKEKLPG and ⅡK segment: (Q/E)K(P/A)G(M/L)LDKIK(A/Q)(K/M)(I/L)PG motif.

**Figure 2 plants-11-02700-f002:**
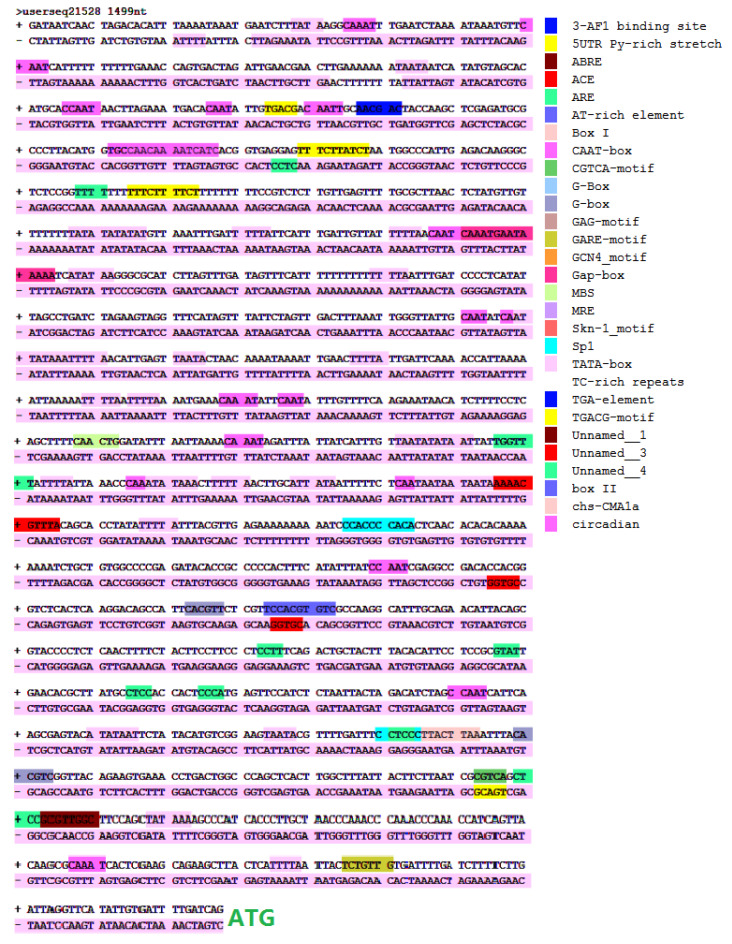
Nucleotide sequence of the PtrDHN-3 promoter (1500 bp). Abiotic stress-related cis-regulatory elements are boxed and labeled with colors.

**Figure 3 plants-11-02700-f003:**
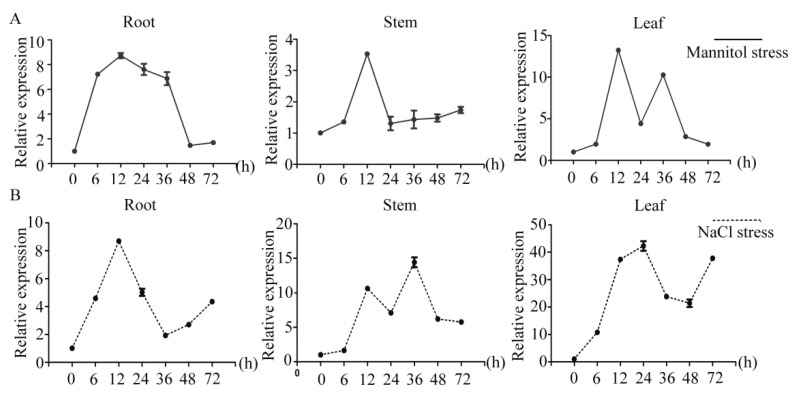
Expression patterns of the *PtrDHN*-3 gene in root, stem and leaf samples from *P. trichocarpa* sampled at 0, 6, 12, 24, 48, and 72 h. Time-course expression patterns of *PtrDHN-3* in response to (**A**) mannitol stress. (**B**) NaCl stress. Expression under control conditions was assigned a value of 1, and then expression under stress treatment was normalized to this level. Error bars indicate ± SD, and each data point represents the mean of three replicate experiments.

**Figure 4 plants-11-02700-f004:**
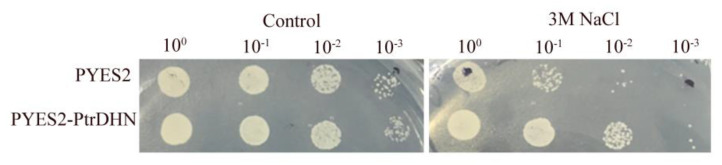
Analysis of salt tolerance in *PtrDHN**-3*-overexpressing and control (transformed with an empty pYES2 vector) yeast cells. Yeast cultures were grown in SC-ura liquid medium containing 2% (*w*/*v*) galactose for 1 d at 30 °C. Concentrations of *PtrDHN**-3*-overexpressing and control yeast cells were adjusted to the same level then identical numbers of cells were resuspended in 3 M NaCl for 2 d. Yeast clones diluted 0, 10, 100, and 1000 times were then spotted onto SC-ura agar plates and incubated at 30 °C for 3 d.

**Figure 5 plants-11-02700-f005:**
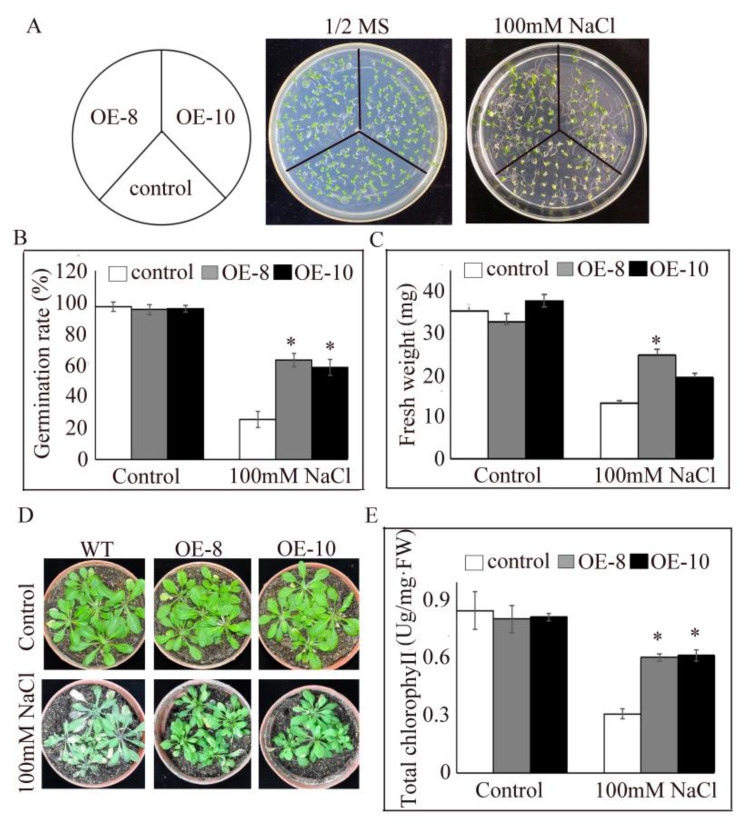
Analysis of salt tolerance in *PtrDHN-3*-overexpressing Arabidopsis plants. (**A**) Seed germination in the transformed and wild-type (WT) lines under control (1/2 MS) and salt stress (100 mM NaCl) conditions. (**B**) Seed germination rates, and analyses of (**C**) fresh weights, (**D**) growth phenotypes, and (**E**) chlorophyll contents. WT and OE Plants were grown on 1/2 MS medium (control) or 1/2 MS medium plus 100 mM NaCl for two weeks for analysis of seed germination rates. For analysis of fresh weights, WT and OE plants germinated on 1/2 MS medium plates for five days were transferred to 1/2 MS medium plates with or without 100 mM NaCl. Measurements were obtained after continuous cultivation for seven days. For observations of growth phenotypes, 4-week-old T3 transgenic and WT Arabidopsis seedlings grown in pots were treated with 100 mM NaCl solution for 10 days then their phenotypes were photographed. The chlorophyll content of leaves was measured. Asterisks indicate a significant difference compared with the WT plants at *p* < 0.05.

**Figure 6 plants-11-02700-f006:**
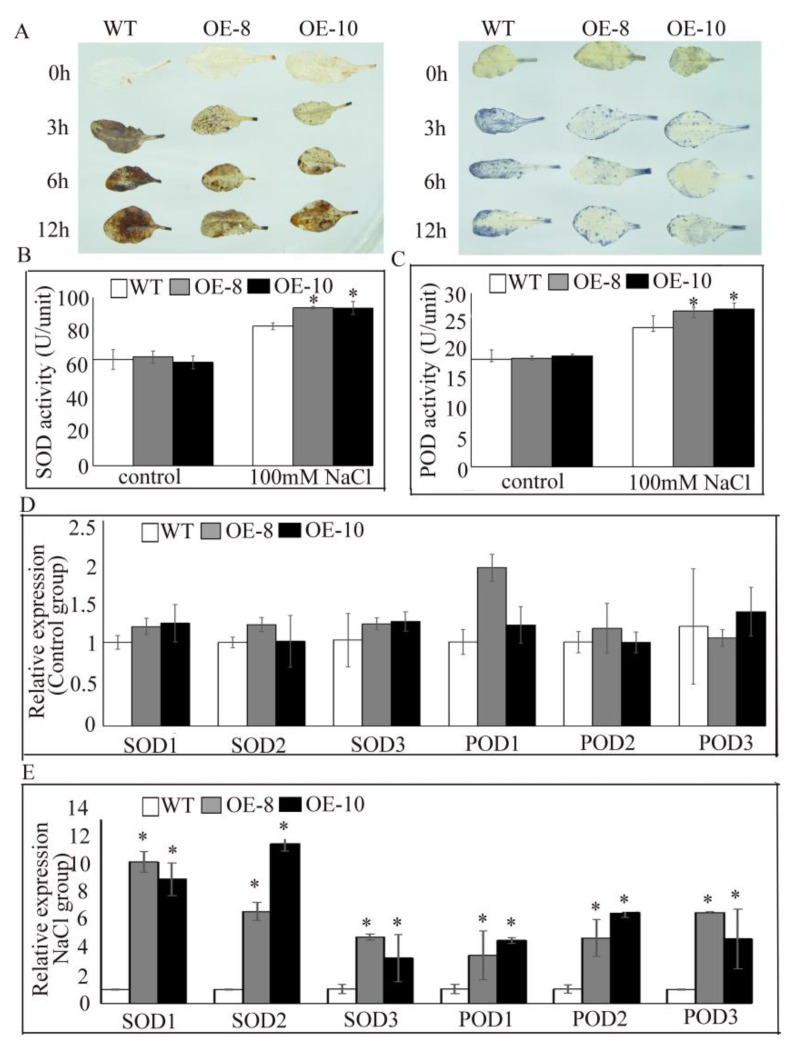
Analysis of ROS scavenging in *PtrDHN-3*-overexpressing Arabidopsis plants. (**A**) DAB and NBT staining were carried out in young leaves of WT and overexpressing plants treated with and without 100 mM NaCl for 0, 3, 6 and 12 h. WT and overexpressing plants were also evaluated for SOD (**B**) and POD (**C**) activity, and for SOD (**D**) and POD (**E**) gene expression. In the expression analysis, results were normalized to those of the Con plants (0 h), then data were log2 transformed. Genbank numbers of the *SODs* and *PODs* are shown in Appendix A. Three independent biological repeats were performed per treatment, and each experiment consisted of at least 9 transformed seedlings. * Indicates a significant difference according to Student’s *t* test at *p* < 0.05 compared with the WT plants.

**Figure 7 plants-11-02700-f007:**
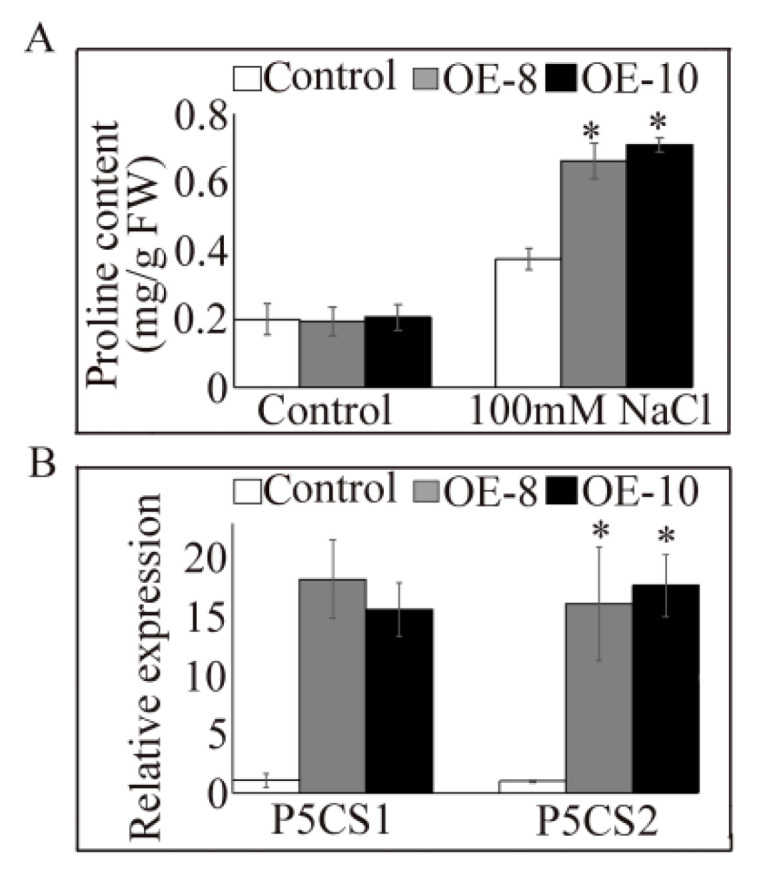
Levels of proline in *PtrDHN-3*-overexpressing Arabidopsis plants. Wild-type (WT) and overexpressing plants were grown in 1/2 MS medium with or without 100 mM NaCl for 48 h and then analyzed. (**A**) Proline contents (**B**) expression of proline biosynthesis-related genes under salt stress. Asterisks indicate a significant difference compared to the control plants at *p* < 0.05.

## Data Availability

Date is contained within the article or Appendix A.

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
