# Peer review of "The Poplar (Populus trichocarpa) Dehydrin Gene PtrDHN-3 Enhances Tolerance to Salt Stress in Arabidopsis"

_plants, 2022, doi:10.3390/plants11202700_

Round 1
Reviewer 1 Report
The study titled “The Poplar (Populus trichocarpa) Dehydrin Gene PtrDHN-3 Enhances Tolerance to Salt Stress in Arabidopsis” by Zhou et al. demonstrated contribution of PtrDHN-3 to salinity tolerance of poplar. The work is a valuable confirmation of previous published results that dehydrin gene plays a fundamental role in plant response and adaptation to abiotic stresses. The manuscript is well written and presented. However, the below issues need to be addressed to improve the submission before publication.
- Gene name should be also italics in the manuscript title.
- One important growth condition is not provided; that is, PPFD in µmol m-2 s-1. This should be given for reproducibility purpose. Also, in lines 360-361, detailed growth conditions are missing, and are not provided in the subsection 2.5 as stated by the authors. This also should be provided.
- The methods of chlorophyll and proline determination should be briefly described.
- Lines 104, 118, and 121: no interpretation of data should be presented in the result section. The same is repeated in other places of the result section. This must be corrected.
Author Response
Dear Editors and Reviewers,
感谢您对我们的手稿的来信和评论,标题为“杨树(杨树毛滴虫)脱水素基因PtrDHN-3增强拟南芥对盐胁迫的耐受性
".这些评论帮助我们改进了手稿,并为未来的研究提供了重要的指导。
我们在逐点回复中处理了编辑和审稿人的意见,并修改了文本以满足植物的要求。我们希望这符合出版要求。主要意见和我们的具体回应详述如下:
审稿人: 1
- 基因名称在手稿标题中也应为斜体。
回应:非常感谢您指出这个问题。这在手稿中得到了纠正和记录。
- 没有提供一个重要的生长条件;也就是说,以微摩尔 m-2 s-1 为单位的 PPFD。这应该是为了可重复性的目的而给出的。此外,在第360-361行中,缺少详细的生长条件,并且如作者所述,在第2.5小节中没有提供。这也应该提供。
回复:感谢您的评论。我们在手稿中添加了生长条件。361-362路和365-364路。
361-362路
然后将它们转移到具有或不具有(对照)100 mM NaCl的1/2 MS培养基板中,并在22°C下用16 h光的光周期和80-120μmol∙m-2∙s-1的光子流动进行包扎。
365-364路
种植方法、养分基质和栽培环境如第4.5小节所述。
- 应简要描述叶绿素和脯氨酸测定的方法。
回应:我们非常感谢您的建议。叶绿素含量根据光瀚(1987)的描述确定[47]。用液氮完全研磨样品并用丙酮处理。测量了超质在663nm、645nm、652nm和470nm波长处的光密度。对于脯氨酸测量,将研磨样品与试剂1和试剂2充分混合,并在水浴后加入试剂3,并在520nm处测量混合物的光密度。(建成生物有限公司,南京,中国)。373-378路。
- 第 104、118 和 121 行:结果部分不应提供对数据的解释。在结果部分的其他位置重复相同的操作。必须纠正这一点。
回应:非常感谢您指出这个问题。结果中删除了对数据的所有解释。

Reviewer 2 Report
In this manuscript from Zhou et al., the authors phylogenetically analyzed the PtrDHN-3 with DHN proteins from other species and cluster PtrDHN-2 into SKn-type subfamily. They found there are several stress-response cis-elements in PtrDHN-3 promoter region. The expression of PtrDHN-3 can be induced to by salt stress in Populus trichocarpa. PtrDHN-3 can increase the tolerance for salt stress when expressed in yeast. Then the authors overexpressed PtrDHN-3 in Arabidopsis and found that PtrDHN-3 can increase SOD and POD activities to scavenge ROS and improves biosynthesis of proline under salt stress, which result in the plant tolerance to salt stress.
However, the design of the research in the manuscript is too simple and the data didn’t provide valuable information in vivo to understand the function of PtrDHN-3 in salt response in Populus trichocarpa. The manuscript was not well written, and some sentences are difficult to understand. So moderate English changes are required. The images in the figures were not well organized with low quality. The following are additional comments:
1. The introduction didn’t provide sufficient background information. The authors mentioned that there are Y segment, K segment and S segment in DHN proteins, but didn’t provide any information about the molecule function of these segments and the mechanism how they work in stress response reported in previously publications.
2. Line 35-36: Please re-organized this sentence. It is hard to understand what it said.
3. Line 46-52: Please re-write these sentences. It seemed that they were from machine translation.
4. The images in figure 1A have low qualities. Please re-create these pictures with high resolution.
5. In figure 1A legends: “Red squares indicate conserved amino acid sequences of S-and K-segment motifs”. But what did “1 K-segment” and “11 K-segment" mean in figure 1B?
6. Figure 3 was not cited in the manuscript, even though the section 2.3 depicted this figure.
7. What are the arrowheads in the gray circles in figure 3 and figure 4?
8. The primer sequence for trActin and AtActin couldn’t be found in the Supplementary Table S1.
Author Response
Dear Editors and Reviewers,
Thank you for your letter and comments on our manuscript titled“The Poplar (Populus trichocarpa) Dehydrin Gene PtrDHN-3 Enhances Tolerance to Salt Stress in Arabidopsis
”. These comments helped us improve our manuscript, and provided important guidance for future research.
We have addressed the editor’s and the reviewers’ comments in a point-by-point response, and revised the text to meet the requirements of plants. We hope this meets the requirements for publication. The main comments and our specific responses are detailed below:
Reviewer: 2
1 The introduction didn’t provide sufficient background information. The authors mentioned that there are Y segment, K segment and S segment in DHN proteins, but didn’t provide any information about the molecule function of these segments and the mechanism how they work in stress response reported in previously publications.
Response:Thank you for your suggestion. Function of K/S/Y segments has been edited and noted in the manuscript. Line 42-46.
Studies indicate that the K-segments are the functional core parts of DHNs that mediate cellular stress tolerance or maintain enzyme activity [49]. S-segment is generally thought to affect the localization or function of DHNs by phosphorylation [50], while the Y-segment is composed of aromatic conserved tyrosine residues whose function remains unclear [51,52]
2 Line 35-36: Please re-organized this sentence. It is hard to understand what it said.
Response:Thank you for pointing this out. This has been corrected and noted in the manuscript. Dehydrin (DHN) is a member of the late embryogenesis abundant protein (LEA) family [6] which accumulate in the late stages of embryogenesis when water content in seeds declines, or in response to various stressors. DHN is clusted into the group II of highly hydrophilic proteins and has been discovered in a large number of plant species [7,8]. Line 35-40.
3 Line 46-52: Please re-write these sentences. It seemed that they were from machine translation.
Response:Thank you for pointing this out. This error has been corrected and noted in the manuscript. Numerous in vitro studies have provided evidence for the potential role of different DHNs in plants. Among the various biochemical activities proposed for these proteins, many suggest a function as membrane or protein stabilizers [18,19], as radical scavengers protecting lipids against peroxidation, or as chaperones preventing stress-induced aggregation of enzymes or proteins [20]. Line 52-57.
4 The images in figure 1A have low qualities. Please re-create these pictures with high resolution.
Response:Thank you for your suggestion. This Fig1 has been replaced in the manuscript.
5 In figure 1A legends: “Red squares indicate conserved amino acid sequences of S-and K-segment motifs”. But what did “1 K-segment” and “11 K-segment" mean in figure 1B?
Response:Thank you for underlining this error. This error has been corrected in the Fig 1. They are the 1 and 2 of the new Roman numerals.
6 Figure 3 was not cited in the manuscript, even though the section 2.3 depicted this figure.
Response:Thank you for pointing this out. This Figure 3 has been cited in the manuscript.
To determine the response of PtrDHN-3 to abiotic stress, qRT-PCR analysis was used to obtain expression profiles under respective treatment with 100mM NaCl and 200mM mannitol. Under mannitol stress (Fig. 3A), the transcripts of PtrDHN-3 were first induced in the roots at 6h followed by the stems and leaves at 12 h, with a peaking at 12h in all tissues.
Meanwhile, treatment with NaCl caused a rapid increase in transcription levels of PtrDHN-3 in the roots, stems and leaves, with a notable increase in the leaves. Levels were highest at 24h, at 41 times that of the control (Fig. 3B).
7 What are the arrowheads in the gray circles in figure 3 and figure 4?
Response:Thank you for pointing this out. These figures has been corrected and noted in the manuscript.
8 The primer sequence for PtrActin and AtActin couldn’t be found in the Supplementary Table S1.
Response:We are really sorry for our fault. We have added the primer sequence for PtrActin and AtActin in the Supplementary Table S1.
|
AtActin2-F |
CCCAGTGTTGTTGGTAGGCCAAGAC |
|
AtActin2-R |
CATAGCGGGAGAGTTAAAGGTCTC |
|
PtrActin2-F |
AACATGGGATTGTTAGCAACTGG |
|
PtrActin2-R |
TCCATCACCAGAATCCAGCACA |

Round 2
Reviewer 2 Report
In the revised manuscript from Zhou et al., the authors made the essential modifications, re-organized the figures and text, and provided important data to address all the concerns raised in the first-round reviews. The manuscript has improved a lot that I think it is ready for publication
Author Response
- Line 356-362: The authors have added a description of their assays for chlorophyll and proline; however, a problem remains. Apparently the authors used a proprietary kit for proline and perhaps the composition of reagent 1, 2 and 3 are not divulged by the manufacturer. As currently written and stating that “…reagents 1 and 2 were mixed thoroughly, and reagent 3 was added after a water bath” is not meaningful to Readers of the manuscript. Authors should at least provide some insight into the method used. For example, the description could be re-stated as follows (authors must provide actual information, as appropriate): “For proline assay, the ground samples were assayed by a colorimetric assay kit based on the ninhydrin reaction (Jiancheng Bio. Co., Nanjing, China) and reaction-product was quantified by its absorbance at 520 nm, calibrated with proline standards..
Response:Thank you very much for pointing out this issue and help us improve the descriptions. This has been corrected and noted in the manuscript.
For proline assay, the ground samples were assayed by a colorimetric assay kit based on the ninhydrin reaction (Jiancheng Bio. Co., Nanjing, China) and reaction-product was quantified by its absorbance at 520 nm, calibrated with proline standards. Line 366-369
- Reviewer 2’s comment 5 was not addressed by the authors in the revised manuscript: In figure 1A legends: ‘Red squares indicate conserved amino acid sequences of S-and K-segment motifs”. But what did “1 K-segment” and “11 K-segment" mean in figure 1B?’ Authors should define what they mean by 1K and 11K.
Response:Thank you for the comment. We have added mean in the manuscript.
K-segment includeⅠK-segment: EKKGIMDIKEKLPG and ⅡK-segment: (Q/E)K(P/A)G(M/L)LDKIK(A/Q)(K/M)(I/L)PG motif. Line100-101
- Line 206: Probably the citation listed to support the statement about cold is incorrect. Probably the authors intended to cite reference 20 by Rorat, et al., 2006. Authors should check all citations in the manuscript because there might be some mix-ups on the numbering. For example, please check these:
Line 209 ref 48 by Lightenthaler?
Line 213 ref 33 by Szekely about Arabidopsis?
Line 231 ref 37 by Kirch et al/ about potato?
Line 233 ref 38 by Wang about Tamarix?
Line 245 ref 41 about Betula and cell walls?
Line 247 ref 33 by Szekely about P5CS?
And many more miss-numbered references throughout the manuscript.
.
Response:Thank you very much for pointing out this issue. All of the references has been corrected and noted in the manuscript.
- Line 35-36: The statement has incorrect use of plurals. Modify as: “Dehydrins (DHNs) are members of the late embryogenesis …”
Response:Thank you very much for pointing out this issue. References has been corrected and noted in the manuscript. Line36.
Dehydrins (DHNs) are members of the late embryogenesis abundant protein (LEA) family
- Figure 1 legend: Add the genus-species names that the abbreviations for the following refer: IpDHN, OesDHN, StDHN, CcDHN and DcDHN.
Response:Thank you for the comments. We have added the genus-species names in the manuscript. IpDHN (Ipomoea pes-caprae L. AQZ36519.1), OesDHN (Olea europaea L. subsp. europaea, var. Sylvestris, AKM95889.1), StDHN (Solanum tuberosum L. AAB53203), CcDHN (Coffea canephora Pierre ex Forehn. ABC68275) and DcDHN (Daucus carota L. BAA82445). Line96-99.
- Line 102: what is “Table 1500. Bp upstream sequence…” Perhaps authors mean to say: “The 1500-bp upstream sequence…”
Response:Really sorry about this error.This has been corrected and noted in the manuscript. Line106 The 1500-bp upstream sequence of the initiator codon ATG in the PtrDHN-3 gene was selected for analysis of cis-acting elements in the gene promoter.
- 第200-202行:句子“野生型(WT)和过表达植物在1/2 MS培养基中生长,有或没有100 mM NaCl,然后分析48小时。似乎未正确放置。它目前位于图 7 的面板 B 的描述之后。但是,它指的是面板A中的控制与盐应力处理,因此,请将此句子移至描述面板B之前的位置。
回应:非常感谢您指出这个问题。这在手稿中得到了纠正和记录。204-205号线。
